# Trends in Outcomes of Major Intracerebral Haemorrhage in a National Cohort of Very Preterm Born Infants in Switzerland

**DOI:** 10.3390/children10081412

**Published:** 2023-08-19

**Authors:** Philip Thwaites, Cornelia Hagmann, Juliane Schneider, Sven M. Schulzke, Sebastian Grunt, Thi Dao Nguyen, Dirk Bassler, Giancarlo Natalucci

**Affiliations:** 1Newborn Research, Department of Neonatology, University Hospital Zurich, University of Zurich, CH-8091 Zurich, Switzerland; p.thwaites39@gmail.com (P.T.);; 2Department of Paediatric and Neonatal Intensive Care, University Children’s Hospital Zurich, CH-8032 Zurich, Switzerland; 3Children’s Research Center, University Children’s Hospital Zurich, CH-8032 Zurich, Switzerland; 4Woman-Mother-Child Department, Clinic of Neonatology, University Hospital Center, University of Lausanne, CH-1011 Lausanne, Switzerland; 5Department of Neonatology, University Children’s Hospital Basel UKBB, University of Basel, CH-4056 Basel, Switzerland; 6Division of Neuropediatrics, Development and Rehabilitation, Department of Paediatrics, Inselspital, Bern University Hospital, University of Bern, CH-3010 Bern, Switzerland; 7Child Development Centre, University Children’s Hospital Zurich, CH-8032 Zurich, Switzerland; 8Family Larsson-Rosenquist Foundation Centre for Neurodevelopment, Growth and Nutrition of the Newborn, University Hospital Zurich, University of Zurich, CH-8091 Zurich, Switzerland

**Keywords:** preterm infant, major intraventricular haemorrhage, periventricular, grade 3 intraventricular haemorrhage, periventricular haemorrhagic infarction, outcome, mortality, neurodevelopment

## Abstract

Background: Major brain lesions, such as grade 3 intraventricular haemorrhage (G3-IVH) and periventricular haemorrhagic infarction (PVHI) are among the main predictors for poor neurodevelopment in preterm infants. In the last decades advancements in neonatal care have led to a general decrease in adverse outcomes. Aim: To assess trends of mortality and neurodevelopmental impairment (NDI) in a recent Swiss cohort of very preterm infants with grade 3 intraventricular haemorrhage (G3-IVH) and periventricular haemorrhagic infarction (PVHI). Methods: In this retrospective population-based cohort study, rates of mortality, and NDI at 2 years corrected age were reported in infants born at 24–29 weeks gestational age (GA) in Switzerland in 2002–2014, with G3-IVH and/or PVHI. Results: Out of 4956 eligible infants, 462 (9%) developed G3-IVH (n = 172) or PVHI (n = 290). The average mortality rates for the two pathologies were 33% (56/172) and 60% (175/290), respectively. In 2002–2014, no change in rates of mortality (G3-IVH, *p* = 0.845; PVHI, *p* = 0.386) or NDI in survivors (G3-IVH, *p* = 0.756; PVHI, *p* = 0.588) were observed, while mean GA decreased (G3-IVH, *p* = 0.020; PVHI, *p* = 0.004). Multivariable regression analysis showed a strong association of G3-IVH and PVHI for both mortality and NDI. Death occurred after withdrawal of care in 81% of cases. Conclusion: In 2002–2014, rates of mortality and NDI in very preterm born infants with major brain lesions did not change. The significant decrease in mean GA and changing hospital policies over this time span may factor into the interpretation of these results.

## 1. Introduction

Despite being characterised by two distinct pathophysiologies, grade 3 intraventricular haemorrhage (G3-IVH) and periventricular haemorrhagic infarction (PVHI) are often grouped together as the most severe forms of intracranial lesion in very preterm infants [1,2] and main predictors for unfavourable outcome [3,4]. G3-IVH is an intraventricular haemorrhage with ventricular enlargement [5] and is therefore considered a further complication of germinal matrix haemorrhage. PVHI appears to be the result of a periventricular venous congestion [5] that leads to venous ischemia and secondary haemorrhage of the white matter. Although PVHI coexists in 10–15% with germinal matrix haemorrhage, it is not considered a parenchymal extension thereof [6]. Both lesions overwhelmingly occur in the first 7 days after birth, complicating the neonatal course of 5–25% of extremely preterm born infants [7], and can be accurately diagnosed using bedside cranial ultrasound (cUS) [8]. G3-IVH and PVHI are associated with neonatal mortality rates of 10–30% and 20–80%, and long-term neurodevelopmental impairment (NDI) rates of up to 50% and 85%, respectively [3,4]. The high variability seen in short- and long-term outcomes following severe brain lesion in the preterm infant seems to be due to various factors including hospital policy, the heterogeneity of the cohorts reported [9], demographics, the presence of outcome relevant comorbidities [10,11] and, especially in the case of PVHI, the characteristics of the brain lesion [2,12]. In the last decades, some groups reported a decrease in adverse outcome in children having suffered these two pathologies [13,14,15]. To investigate the trend over time of the incidence of these morbidities and their long-term consequences, we performed a population-based analysis of a Swiss cohort of very preterm infants born in 2002–2014. We hypothesised that short- and long-term outcomes of very preterm children with G3-IVH or PVHI have improved over the study period.

## 2. Methods

### 2.1. Study Design and Population

This was a retrospective population-based cohort study investigating mortality and neurodevelopmental outcome at 2 years of corrected age of very preterm infants live born in Switzerland from 1 January 2002 to 31 December 2014 at a gestational age (GA) of 24 to 29 weeks, who developed G3-IVH or PVHI as diagnosed by cUS. Infants born at less than 24 weeks of gestation were not included since the publication of the 2002 Swiss guidelines for the care of infants between 22 and 26 gestational weeks may have affected the decision to provide intensive care in the delivery room for infants born at the limits of viability [16]. Baseline neonatal characteristics, including cUS diagnoses, demographics and follow-up data were extracted from the prospective Swiss prematurity registry of the Swiss Neonatal Network and Follow-up Group [17]. Exclusion criteria were the following: GA below 24 weeks, genetic syndromes, inborn metabolic disorders affecting life expectancy or neurodevelopment, major congenital malformations requiring surgical correction or potentially affecting neurodevelopmental outcome. Data collection and evaluation for the present study were approved by the institutional ethical review boards (KEK-ZH-Nr 2014-0552; BASEC-Nr. 2018-00231) and by the Swiss Federal Commission for Privacy Protection in Medical Research. Participating centres were obliged to inform parents about the scientific use of anonymized data.

### 2.2. Cranial Ultrasound

During the neonatal period, the local attending neonatologist or paediatric radiologist performed cUS examinations at minimum on day 1, 3, and 7 of life and repeated weekly or biweekly until hospital discharge. The main findings of the cUS examinations were also entered in the prospective national prematurity registry at infant’s discharge from the hospital and extracted for this study. G3-IVH and PVHI were defined according to Volpe et al. [18].

### 2.3. Neonatal and Sociodemographic Baseline

Neonatal and sociodemographic baseline GA was the best estimate available from the obstetric measurements based on the last menstrual period or on prenatal ultrasound findings recorded in the maternal chart. Baseline neonatal characteristics were previously described [11]. The socioeconomic status (SES) of the study subjects’ parents was estimated using a reliable and valid 12-point scale [19] based on maternal education and paternal occupation with 2 being the lowest and 12 the highest level.

### 2.4. Neurodevelopmental Outcome

Experienced developmental paediatricians, paediatric neurologists or neuropsychologists carried out the neurodevelopmental assessment of the infants during a routine follow-up examination in one of the 16 Swiss Follow-Up Group Centres at corrected age of 2 years. The assessment consisted of a clinical examination, a structured neurological and a standardised developmental assessment. During the study period, the assessments used were the Bayley Scales of Infant Development, 2^nd^ edition (BSID-II) [20] and, following its introduction, the Bayley Scales of Infant and Toddler Development, 3^rd^ edition (Bayley-III) [21]. Up to 2006, some children were tested using the Griffiths Mental Development Scales-Revised (GMDS) [22]. Children with significant disability preventing completion of the development test were assigned a development score of 54 (1 below −3SD). Cerebral palsy was defined according to Rosenbaum et al. [23] and was graded according to the Gross Motor Function Classification System (GMFCS) for children aged ≤2 years [24]. Vision and hearing were assessed by either direct examination or caregiver report. 

### 2.5. Outcome Definition

The outcomes of the present study were mortality, NDI, and favourable outcome at 2 years of corrected age. NDI was defined as one of the following: mental or motor development index below 70 (−2SD) on the BSID-II, cognitive or motor composite score below 85 (−1SD, see below) on the Bayley-III, a global score on the GMDS below 76 (−2SD), cerebral palsy with GMFCS above 1, the absence of useful hearing, blindness, or perception of light only. According to recent literature, Bayley-III scores below 85 were considered as equivalent to indices below 70 (−2SD) in the BSID-II [25]. Favourable outcome was defined as absence of any of the above.

### 2.6. Statistical Analysis

Baseline characteristics of infants with G3-IVH and with PVHI were compared separately with those of children with normal cUS findings using the independent Student’s t- and χ^2^ tests as appropriate. The same procedure was used to compare baseline characteristics of infants with major brain lesion with and without 2-year follow-up. Descriptive statistics were used for the following outcomes: mortality, NDI, and favourable outcome. The association of G3-IVH and PVHI with the measured outcomes of death and NDI among survivors who attended the follow-up visit was calculated with univariable and multivariable logistic regression models that were adjusted for known neurodevelopmental risk factors including GA, sex, birth weight z-score, multiple birth, complete antenatal steroids course, necrotising enterocolitis Bell’s stage 2 or higher, sepsis, retinopathy of prematurity stage 3 or higher, moderate to severe bronchopulmonary dysplasia, family SES. Associations are given as odds ratio (OR) with 95% confidence interval (95% CI). To determine temporal changes in the distribution of outcomes, we used the Cochran–Armitage test for trends in proportions. The statistical significance threshold was defined as *p* < 0.05, and testing was two-sided. Statistical analyses were performed with R release 2.13.0 (R Core Team (2021). R: A language and environment for statistical computing. R Foundation for Statistical Computing, Vienna, Austria).

## 3. Results

### 3.1. Study Population

From 2002 to 2014, 5224 children with a GA of 24 to 29 weeks were live born in Switzerland. Among them, 4956 fulfilled the inclusion criteria and 462 (9%) were diagnosed with G3-IVH (n = 172, 3%) or PVHI (n = 290, 6%). A total of 3296 infants (67%) with normal cUS findings composed the control group for outcome comparisons (Figure 1). Infants with G3-IVH displayed lower GA and 10 min Apgar score, and higher rates of male sex, necrotising enterocolitis, sepsis, and patent ductus arteriosus than infants with normal cUS findings. Compared to infants with normal cUS findings, infants with PVHI displayed the following: lower GA, birth weight z-score, arterial cord pH, and 10 min Apgar score; lower rates of completed antenatal steroids administration and caesarean section; higher rates of male sex, necrotising enterocolitis, bronchopulmonary dysplasia, and patent ductus arteriosus (Table 1).

### 3.2. Trends in Outcomes of G3-IVH and PVHI

Figure 2 shows trends in mortality, NDI, and favourable outcome over the period from 2002 to 2014 in infants with G3-IVH (shown in Figure 2a) and PVHI (shown in Figure 2b), respectively. According to the Cochran–Armitage test (Appendix A), there was no significant change in rates of G3-IVH and of PVHI, of mortality (G3-IVH, *p* = 0.845; PVHI, *p* = 0.386), NDI (G3-IVH, *p* = 0.757; PVHI, *p* = 0.588), and favourable outcome (G3-IVH, *p* = 0.516; PVHI, *p* = 0.265), while GA decreased significantly in both groups (G3-IVH, *p* = 0.020; PVHI, *p* = 0.004). Over the whole period (2002–2014), the average mortality rates were 33% (56/172) and 60% (175/290) in infants with G3-IVH and with PVHI, respectively. Death occurred after withdrawal of care in 81% of cases (45/56 in G3-IVH and 141/175 in PVHI).

Neurodevelopmental follow-up assessment was performed in 104/116 (90%) and in 92/115 (80%) of surviving infants with G3-IVH and PVHI, at a mean (SD) corrected age of 22.2 (3.5) and 22.8 (3.4) months, respectively. Baseline characteristics of infants visited at follow-up (196/231, 85%) and of those lost to follow-up (35/231) were similar (Appendix A). Over the whole period, among infants assessed for the 2-year follow-up, 34% (35/104) of infants with G3-IVH and 53% (49/93) of infants with PVHI showed NDI, respectively.

### 3.3. Association between Major Brain Lesions and the Outcomes of Death and NDI

References for the calculation of the ORs in the outcomes assessed in infants with major brain lesions were defined by the group of infants with normal cUS who displayed an overall mortality rate of 15% (493/3296) and an NDI rate of 15% among survivors tested at a mean (SD) age of 22.7 (3.0) months. The ORs for the outcome of death were significant at 2.7 (95% CI 2.0 to 3.8) in infants with G3-IVH in the univariable comparison with infants with normal cUS findings and 3.3 (1.9 to 5.6) in the multivariable analysis, while they were 8.6 (6.7 to 11.1) and 9.5 (6.1 to 14.8) for infants with PVHI, respectively. When restricting the dataset to surviving infants, the ORs for NDI were 3.0 (1.9 to 4.5) in infants with G3-IVH in the univariable and 2.5 (1.5 to 4.0) in the multivariable comparison with infants with normal cUS findings. In children with PVHI they were 6.5 (4.2 to 10.0) and 6.7 (4.1 to 10.9), respectively (Table 2).

## 4. Discussion

This study provides representative outcome data on mortality, NDI, and favourable outcome at a corrected age of 2 years following G3-IVH and PVHI in a recent population-based cohort of infants born in Switzerland. The findings show a nearby constant mortality rate over the 13-year observation period in infants who developed either of these pathologies with an average mortality of approximatively 1/3 in infants with G3-IVH and 2/3 in infants with PVHI, respectively, which occurred after withdrawal of care in 4/5 cases.

The present data contrast with the study hypothesis and with previous studies where mortality of infants with major brain lesion was shown to have decreased over the last decades [14,15]. Since low GA is a known predictor of adverse neonatal outcome [2,10], the significant decrease in the mean GA identified from 2002 to 2014 in the infants studied might partially explain these findings. However, other factors may be relevant. First, infants with lower GA might have suffered higher rates of the most severe forms of major brain lesion (e.g., rapidly extensive and bilateral lesions) or from some severe complications such as post haemorrhagic ventricular dilatation. In the literature, G3-IVH and PVHI are often defined together as a composite predictor of adverse outcome without any detailed description of the extent and location of the brain lesion. However, some studies have shown that the characteristics of the bleeds such as size, laterality, and midline shift are distinctly associated with neonatal and neurodevelopmental outcomes [3,26,27,28,29]. Second, since the majority of deceased infants died after withdrawal of care, policies regarding the continuation of care in very preterm born infants with major brain lesions may have affected the lack of change in mortality rates over time. Though better long-term outcomes have been reported in some studies [13,14,15], it appears that the attitude towards care withdrawal in very preterm infants with major brain injury has not evolved during this period and in the Swiss perinatal centres studied. It might be possible that the decrease in the GA of the population studied might have influenced decisions to discontinue neonatal care, due to presumption of adverse outcome, within the group of participants with major brain lesions.

G3-IVH and PVHI in the present cohort were also shown to be strong predictors for poor long-term outcome, with approximatively 1/3 and 1/2 of infants being diagnosed with NDI in the 2-year follow-up, respectively. However, the high rate of discontinuation of neonatal care limits the interpretation of any developmental outcome measure due to its premature truncation. In fact, the absence of a trend over time for neurodevelopmental outcomes at 2 years of age in survivors after major brain lesion may be due to the selective process linked to the withdrawal of care policies of the neonatal centres. Survival rate might have been higher in infants with higher GA, less severe forms of major brain lesions, and less comorbidities. The strengths of the present study are its large sized, well-defined group of infants with major brain lesions from a population-based cohort with similar therapeutic approaches and standards of care. Additionally, the outcomes measured were all based on validated standards.

The following limitations that reduce the generalisability of the findings need to be addressed. First, an information bias may have been caused by the lack of magnetic resonance imaging data that capture the full extent of brain lesions. The authors acknowledge also that important imaging data characterising concurrent lesions such as cerebellar haemorrhages or diffuse white matter injury as well as post-haemorrhagic complications such as dilatation and inflammation are lacking in the present study. The analysis of the association between lesion characteristics and neurodevelopmental outcome in a subgroup of this cohort (where cUS imaging data is available) is ongoing. Second, the low numbers of events (G3-IVH or PVHI) observed per year could have limited the power of the analysis of trend over time. Third, 15% of survivors after major brain lesion were not tested at age 2 years and were thus excluded from the analysis of outcome trends over time and the regression models. Although infants lost to follow-up displayed similar baseline characteristics to those of the study group, this loss of information may have caused an underestimation of the reported overall outcome. Lastly, the neurodevelopmental outcome has been assessed at a relative early age leaving further important outcome information such as schooling or social integration of the studied infants unknown. However, the predictive validity of early assessments for long-term neurodevelopment is acceptable when focusing exclusively on infants with major disabilities as in the present study and on proportions of preterm born children with severe and moderate impairment remain stable over childhood [30].

In conclusion, no significant changes in mortality or neurodevelopmental outcome at 2 years of age could be shown in very preterm infants with G3-IVH or PVHI between 2002 and 2014. This may be due to a decreasing mean GA, severity of lesions, and also to the low number of events or changes in hospital policy regarding withdrawal of care over the years assessed. The present study highlights the need for future investigations that focus not only on the outcome of survivors after extreme prematurity and major brain lesion but also on the local policies and the reasons of withdrawal of care. Additionally, predictors of neurodevelopment in this population, e.g., the characteristics of the brain lesions such as extension and localization should be specifically refined.

## Figures and Tables

**Figure 1 children-10-01412-f001:**
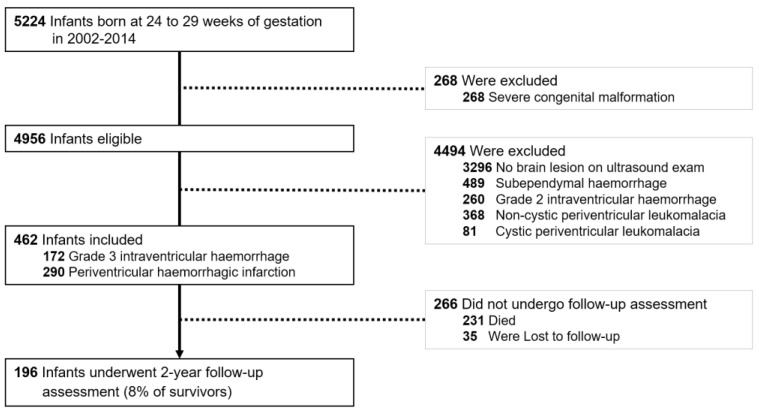
Study flow chart.

**Figure 2 children-10-01412-f002:**
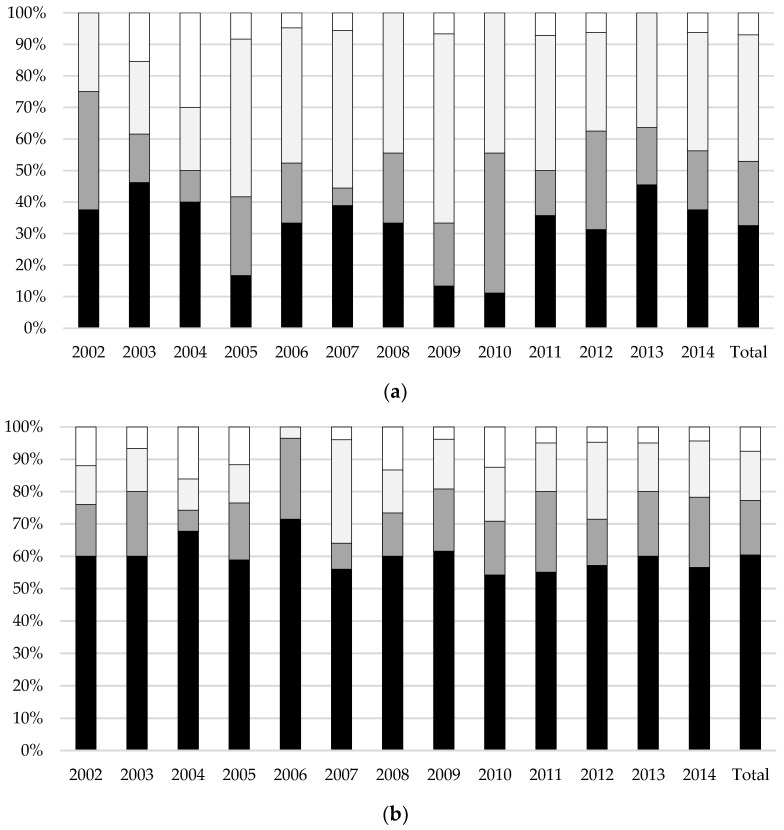
Trends in rates of mortality (black), neurodevelopmental impairment (dark grey), favourable outcome (light grey), and lost to follow-up (white) of infants with grade 3 intraventricular haemorrhage (**a**) and periventricular haemorrhagic infarction (**b**), respectively, over the period from 2002 to 2014.

**Table 1 children-10-01412-t001:** Neonatal baseline characteristics and demographics of the study cohort.

Mean (SD), N (%)	G3-IVH N = 172	Comparison with Normal cUS OR (95% CI)	PVHI N = 290	Comparison with Normal cUS OR (95% CI)	Normal cUS N = 3296 (Reference)
Male sex	103 (60%)	1.37 (1.00 to 1.87) *	184 (63%)	1.59 (1.24 to 2.04) ***	1719 (52%)
Gestational age (weeks)	26.9 (1.5)	−0.7 (−0.9 to −0.4) ***	26.5 (1.6)	−1.1 (−1.3 to −0.9) ***	27.6 (1.7)
Multiple status	47 (27%)	0.84 (0.59 to 1.18)	85 (29%)	0.92 (0.71 to 1.20)	1021 (31%)
Birth weight (grams)	940 (259)	−45 (−89 to −2) *	913 (256)	−72 (−106 to −38) ***	985 (285)
z-score	−0.01 (0.84)	0.13 (−0.00 to 0.26)	0.04 (0.81)	0.19 (0.08 to 0.29) ***	−0.14 (0.90)
Small for gestational age	12 (7%)	0.66 (0.36 to 1.19)	20 (7%)	0.65 (0.41 to 1.03)	338 (10%)
Completed antenatal steroids course	108/170 (63%)	0.78 (0.57 to 1.08)	138/283 (49%)	0.43 (0.33 to 0.54) ***	2173 (66%)
Chorioamnionitis	15/171 (9%)	0.97 (0.56 to 1.66)	32/289 (11%)	1.25 (0.85 to 1.84)	297/3285 (9%)
Caesarean section	130/171 (76%)	0.80 (0.56 to 1.15)	184/289 (64%)	0.45 (0.35 to 0.57) ***	2582/3237 (80%)
Arterial cord pH	7.28 (0.18)	−0.02 (−0.04 to 0.00)	7.28 (0.15)	−0.02 (−0.04 to −0.01) **	7.30 (0.12)
10’ Apgar score	7.1 (1.9)	−0.6 (−0.9 to −0.3) ***	7.3 (1.8)	−0.4 (−0.7 to −0.2) **	7.8 (2.1)
NEC n (%)	10 (6%)	2.69 (1.36 to 5.30) **	14 (5%)	2.21 (1.23 to 3.96) **	74 (2%)
Sepsis n (%)	41 (24%)	2.00 (1.39 to 2.88) ***	39 (13%)	0.99 (0.70 to 1.41)	446 (13%)
Oxygen after 36 Weeks PMA n (%)	31 (18%)	1.48 (0.99 to 2.21)	22 (8%)	0.55 (0.35 to 0.86) **	414 (12%)
ROP n (%)	3 (1.7%)	0.77 (0.24 to 2.48)	4 (1%)	0.61 (0.22 to 1.68)	74 (2%)
Patent arterial duct	91 (53%)	2.53 (1.86 to 3.44) ***	136 (47%)	1.99 (1.56 to 2.53) ***	1014 (31%)
Length of hospital stay (days)	56.8 (41.6)	2.4 (−3.2 to 8.0)	35.1 (39.7)	−19.3 (−23.7 to −14.9) ***	54.5 (35.9)
Family socioeconomic status	6.2 (2.6)	0.1 (−0.4 to 0.5)	6.2 (2.7)	0.1 (−0.3 to 0.5)	6.1 (2.6)

cUS, cranial ultrasound; G3-IVH, grade 3 intraventricular haemorrhage; PVHI, periventricular haemorrhagic infarction; NEC, necrotising enterocolitis; ROP, retinopathy of prematurity; PMA, postmenstrual age; independent Student’s t- and χ^2^ tests as appropriate. * *p* < 0.05; ** *p* < 0.01; *** *p* < 0.001.

**Table 2 children-10-01412-t002:** Outcomes of very preterm infants with grade 3 intraventricular haemorrhage (G3-IVH) and periventricular haemorrhagic infarction (PVHI).

N (%)	G3-IVH	OR (95% CI) ^1^	Adjusted † OR (95% CI) ^1^	PVHI	OR (95% CI) ^2^	Adjusted † OR (95% CI) ^2^	Normal cUS (Reference)
**Total live births**	172	N.A.	N.A.	290	N.A.	N.A.	3296
**Death**	56 (33%)	2.75 (1.97 to 3.83) ***	3.29 (1.94 to 5.57) ***	175 (60%)	8.65 (6.71 to 11.15) ***	9.47 (6.06 to 14.81) ***	493 (15%)
**Follow-up among** **survivors**	104/116	N.A.	N.A.	92/115	N.A.	N.A.	1776/2803
**Neurodevelopmental impairment**	35/104 (34% of tested)	2.96 (1.93 to 4.53) ***	2.46 (1.50 to 4.03) ***	49/92 (53% of tested)	6.50 (4.24 to 9.97) ***	6.68 (4.08 to 10.92) ***	259/1776 (15% of tested)
**Cognitive development** **< - 2 SD**	25/104 (24% of tested)	3.46 (2.14 to 5.59) ***	2.76 (1.56 to 4.89) ***	31/92 (34% of tested)	5.46 (3.44 to 8.68) ***	4.85 (2.81 to 8.36) ***	149/1776 (8% of tested)
**Motor development** **< - 2 SD**	20/104 (19% of tested)	2.42 (1.45 to 4.05) **	1.74 (0.94 to 3.25)	37/92 (40% of tested)	6.72 (4.30 to 10.50) ***	7.52 (4.50 to 12.55) ***	159/1776 (9% of tested)
**Cerebral palsy**	21/104 (20% of tested)	7.49 (4.34 to 12.92) ***	6.25 (3.31 to 11.80) ***	47/92 (51% of tested)	30.92 (19.03 to 50.24) ***	33.13 (18.82 to 58.32) ***	58/1776 (3% of tested)
**GMFCS > 1**	9/104 (5% of tested)	10.43 (4.49 to 24.21) ***	6.47 (2.29 to 18.29) ***	15/92 (16% of tested)	21.27 (10.10 to 44.36) ***	19.27 (7.94 to 46.77) ***	16/1776 (0.5% of tested)
**Severe visual** **problems**	0/104	-	-	3/92 (3% of tested)	11.81 (2.78 to 50.21) ***	20.15 (3.67 to 110.71) **	5/1776 (0.3% of tested)
**Severe hearing** **problems**	2/104 (2% of tested)	6.95 (1.33 to 36.25) **	3.39 (0.52 to 22.11)	3/92 (3% of tested)	11.81 (2.78 to 50.21) ***	15.01 (2.88 to 78.21) **	5/1776 (0.3% of tested)
**Ongoing therapy**	21/104 (20% of tested)	2.83 (1.70 to 4.70) ***	3.12 (1.79 to 5.44) ***	30/92 (33% of tested)	5.32 (3.34 to 8.48) ***	7.37 (4.36 to 12.47) ***	146/1777 (8% of tested)

cUS, cranial ultrasound; G3-IVH, grade 3 intraventricular haemorrhage; ^1^, comparison between normal cUS and G3-IVH; ^2^, comparison between normal cUS and PVHI; †, adjusted for gestational age, sex, birth weight z-score, multiple birth, complete antenatal steroids course, necrotising enterocolitis Bell’s stage two or higher, sepsis, retinopathy of prematurity stage 3 or higher, moderate to severe bronchopulmonary dysplasia, family socioeconomic status; GMFCS, gross motor function classification system; * *p* < 0.05; ** *p* < 0.01; *** *p* < 0.001.

## Data Availability

The data that support the findings of this study are available on request from the corresponding author. The data are not publicly available due to ethical restrictions.

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
