# Peer review of "Trends in Outcomes of Major Intracerebral Haemorrhage in a National Cohort of Very Preterm Born Infants in Switzerland"

_children, 2023, doi:10.3390/children10081412_

Round 1
Reviewer 1 Report
Dear Authors,
I had the proviledge to review your manuscript entitled "Trends in outcomes of major intracerebral haemorrhage in a national cohort of very preterm born infants in Switzerland."
This is an interesting work examining clinical impact and outcome trends of intracerebral hemorrhage of prematures in Switzerland.
The systematic collection of data from a homogeneous health setting is a double edged sword, contributing to standardization of results while introducing a systemic bias. Nonetheless, this issue is cleraly stated and discussed in the manuscript.
Overall, this work concisely and elegantly reports collected data and Authors perfectly put them in the context of previously available knowledge.
Neurodevelopmental outcome in prematures with intracerebral hemorrhage is significantly conditioned by hydrocephalus and its complications. In fact, cerebrospinal fluid infection is a major player in contributing to dismal outcomes. It is not clear why the Authors omitted this piece of information. This should be added to their data analysis or it should be discussed as a serious limitation of this work.
Finally, I would suggest a minor spelling check of the manuscript.
Author Response
We thank Reviewer 1 for taking the time to look at our manuscript.
Point 1: Neurodevelopmental outcome in prematures with intracerebral hemorrhage is significantly conditioned by hydrocephalus and its complications. In fact, cerebrospinal fluid infection is a major player in contributing to dismal outcomes. It is not clear why the Authors omitted this piece of information. This should be added to their data analysis or it should be discussed as a serious limitation of this work.
Reply 1: We already discussed the importance of post-haemorrhagic dilatation (hydrocephalus) in the first manuscript version and very agree with its relevance. We reformulated the limitation paragraph accordingly to include the aforementioned conditions. Unfortunately the information on additional or concurrent lesions is not present in the registry, at least for the time period we looked at. As we mentioned in the text, precise analysis of the association between lesion characteristics and neurodevelopmental outcome in a subgroup of this cohort (where imaging data is available) is ongoing. Please see page 7, line 242 - 247.
Finally, I would suggest a minor spelling check of the manuscript.
Reply 2: The text was proof read by a native english (British) speaker, could this explain the spelling issues referred to?
Reviewer 2 Report
This study is a retrospective analysis of data on survival and clinical outcome for the very preterm born infants (24-29 GA) who had intracerebral hemorrhage (IVH or PVHI) in the perinatal and early postnatal period. The manuscript raises questions about the lack of positive dynamics in the survival of such children despite the improvement in the quality of medical care within the time range from 2002 to 2014, as well as the justification and possible reasons for premature termination of care for such children, which in 80% of cases leads to their death. I would like to note the good presentation of the results obtained.
While reading the manuscript, several comments arose:
1. Are there data from similar studies in other countries? how do they compare with the results of this study? It would be interesting to see this kind of information in the discussion.
2. What is the opinion of the authors, is it justified to refuse assistance to such children, taking into account the fact that new promising approaches to the treatment of such pathologies are being developed, including stem cells and their secretion products?
Author Response
We thank Reviewer 2 for taking the time to look at our manuscript and for the interesting questions posed. Especially, with regard to point 2, it was our intention to make the reader aware of this issue.
Point 1. Are there data from similar studies in other countries? how do they compare with the results of this study? It would be interesting to see this kind of information in the discussion.
Reply 1: Data from similar studies regarding short (mortality) and long (NDI) term outcome of infants are referenced in the second paragraph of the discussion section. Additionally, as mentioned in the introduction, it is difficult to compare outcomes from the literature, due to varying hospital policies, heterogeneity of cohorts reported, demographics, the presence of outcome relevant comorbidities and for PVHI bleed characteristics.
Point 2: What is the opinion of the authors, is it justified to refuse assistance to such children, taking into account the fact that new promising approaches to the treatment of such pathologies are being developed, including stem cells and their secretion products?
Reply 2: Given the fact we provide evidence that in some cases neurodevelopmental outcome may not be dramatically affected in survivors after major brain bleeding (2/3 of infants with G3-IVH and 1/2 of infants with PVHI showed a favorable outcome), as well as the novel neuroprotective approaches mentioned by the reviewer (we include also the pharmacological approach), the authors believe that going forward withdrawal of care decisions need to be thoroughly scrutinized.
Reviewer 3 Report
The goal of the manuscript by Thwaites et al. is to assess the trends of mortality and NDI in a Swiss cohort of very preterm infants with G3-IVH and PVHI. It is a retrospective population-based cohort study comprising 4956 eligible infants from January 2002 until December 2014.
The Introduction section contains an adequate amount of information on the state-of-the-art. The Materials and Methods section is well conceptualized, with clear inclusion and exclusion criteria, a detailed overview of the tests employed for evaluating neurodevelopmental outcome and appropriate statistical tests outlined. The Results section contains a good graphical depiction of the study flow chart (Fig. 1) and trends observed (Fig. 2), with a detailed breakdown of the neonatal baseline characteristics and demographics of the study cohort (Tab. 1) and observed outcomes (Tab. 2).
The Discussion section is well written, with clear reference to the differences observed with respect to the state of the art, including clear reference to potential limitations. As such, I recommend the publication of this manuscript in Children.
Author Response
We thank Reviewer 3 for this positive feedback.